# The Functional Neural Process

**Christos Louizos**
University of Amsterdam
TNO Intelligent Imaging
c.louizos@uva.nl

**Xiahan Shi**
Bosch Center for Artificial Intelligence
UvA-Bosch Delta Lab
xiahan.shi@de.bosch.com

**Klamer Schutte**
TNO Intelligent Imaging
klamer.schutter@tno.nl

**Max Welling**
University of Amsterdam
Qualcomm
m.welling@uva.nl

## Abstract

We present a new family of exchangeable stochastic processes, the Functional Neural Processes (FNPs). FNPs model distributions over functions by learning a graph of dependencies on top of latent representations of the points in the given dataset. In doing so, they define a Bayesian model without explicitly positing a prior distribution over latent global parameters; they instead adopt priors over the relational structure of the given dataset, a task that is much simpler. We show how we can learn such models from data, demonstrate that they are scalable to large datasets through mini-batch optimization and describe how we can make predictions for new points via their posterior predictive distribution. We experimentally evaluate FNPs on the tasks of toy regression and image classification and show that, when compared to baselines that employ global latent parameters, they offer both competitive predictions as well as more robust uncertainty estimates.

## 1 Introduction

Neural networks are a prevalent paradigm for approximating functions of almost any kind. Their highly flexible parametric form coupled with large amounts of data allows for accurate modelling of the underlying task, a fact that usually leads to state of the art prediction performance. While predictive performance is definitely an important aspect, in a lot of safety critical applications, such as self-driving cars, we also require accurate uncertainty estimates about the predictions.

Bayesian neural networks [33, 37, 15, 5] have been an attempt at imbuing neural networks with the ability to model uncertainty; they posit a prior distribution over the weights of the network and through inference they can represent their uncertainty in the posterior distribution. Nevertheless, for such complex models, the choice of the prior is quite difficult since understanding the interactions of the parameters with the data is a non-trivial task. As a result, priors are usually employed for computational convenience and tractability. Furthermore, inference over the weights of a neural network can be a daunting task due to the high dimensionality and posterior complexity [31, 44].

An alternative way that can "bypass" the aforementioned issues is that of adopting a stochastic process [25]. They posit distributions over functions, e.g. neural networks, directly, without the necessity of adopting prior distributions over global parameters, such as the neural network weights. Gaussian processes [41] (GPs) is a prime example of a stochastic process; they can encode any inductive bias in the form of a covariance structure among the datapoints in the given dataset, a more intuitive modelling task than positing priors over weights. Furthermore, for vanilla GPs, posterior inference is much simpler. Despite these advantages, they also have two main limitations: 1) the

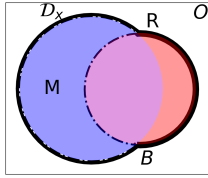

Figure 1: Venn diagram of the sets used in this work. The blue is the training inputs $\mathcal{D}_x$, the red is the reference set $R$ and the parts enclosed in the dashed and solid lines are $M$, the training points not in $R$, and $B$, the union of the training points and $R$. The white background corresponds to $O$, the complement of $R$.

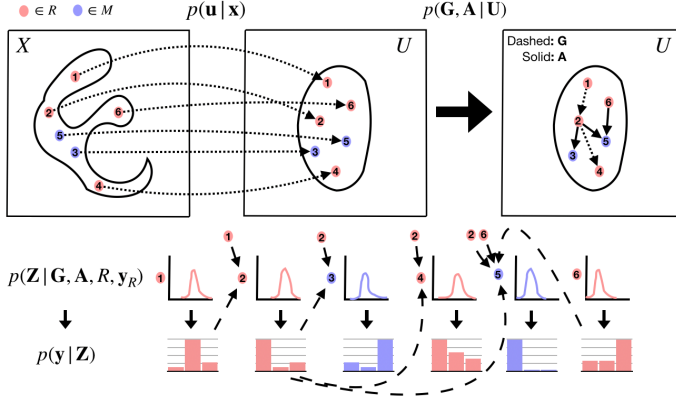

Figure 2: The Functional Neural Process (FNP) model. We embed the inputs (dots) from a complicated domain $\mathcal{X}$ to a simpler domain $U$ where we then sample directed graphs of dependencies among them, $\mathbf{G}, \mathbf{A}$. Conditioned on those graphs, we use the parents from the reference set $R$ as well as their labels $\mathbf{y}_R$ to parameterize a latent variable $\mathbf{z}_i$ that is used to predict the target $y_i$. Each of the points has a specific number id for clarity.

underlying model is not very flexible for high dimensional problems and 2) training and inference is quite costly since it generally scales cubically with the size of the dataset.

Given the aforementioned limitations of GPs, one might seek a more general way to parametrize stochastic processes that can bypass these issues. To this end, we present our main contribution, *Functional Neural Processes* (FNPs), a family of exchangeable stochastic processes that posit distributions over functions in a way that combines the properties of neural networks and stochastic processes. We show that, in contrast to prior literature such as Neural Processes (NPs) [14], FNPs do not require explicit global latent variables in their construction, but they rather operate by building a graph of dependencies among local latent variables, reminiscing more of autoencoder type of latent variable models [24, 42]. We further show that we can exploit the local latent variable structure in a way that allows us to easily encode inductive biases and illustrate one particular instance of this ability by designing an FNP model that behaves similarly to a GP with an RBF kernel. Furthermore, we demonstrate that FNPs are scalable to large datasets, as they can facilitate for minibatch gradient optimization of their parameters, and have a simple to evaluate and sample posterior predictive distribution. Finally, we evaluate FNPs on toy regression and image classification tasks and show that they can obtain competitive performance and more robust uncertainty estimates. We have open sourced an implementation of FNPs for both classification and regression along with example usages at https://github.com/AMLab-Amsterdam/FNP.

## 2 The Functional Neural Process

For the following we assume that we are operating in the supervised learning setup, where we are given tuples of points $(\mathbf{x}, y)$, with $\mathbf{x} \in \mathcal{X}$ being the input covariates and $y \in \mathcal{Y}$ being the given label. Let $\mathcal{D} = \{(\mathbf{x}_1, y_1) \ldots, (\mathbf{x}_N, y_N)\}$ be a sequence of $N$ observed datapoints. We are interested in constructing a stochastic process that can bypass the limitations of GPs and can offer the predictive capabilities of neural networks. There are two necessary conditions that have to be satisfied during the construction of such a model: exchangeability and consistency [25]. An exchangeable distribution over $\mathcal{D}$ is a joint probability over these elements that is invariant to permutations of these points, i.e.

$$p(y_{1:N}|\mathbf{x}_{1:N}) = p(y_{\sigma(1:N)}|\mathbf{x}_{\sigma(1:N)}), \tag{1}$$

where $\sigma(\cdot)$ corresponds to the permutation function. Consistency refers to the phenomenon that the probability defined on an observed sequence of points $\{(\mathbf{x}_1, y_1), \ldots, (\mathbf{x}_n, y_n)\}$, $p_n(\cdot)$, is the same as the probability defined on an extended sequence $\{(\mathbf{x}_1, y_1), \ldots, (\mathbf{x}_n, y_n), \ldots, (\mathbf{x}_{n+m}, y_{n+m})\}$,

$p_{n+m}(\cdot)$, when we marginalize over the new points:

$$p_n(y_{1:n}|\mathbf{x}_{1:n}) = \int p_{n+m}(y_{1:n+m}|\mathbf{x}_{1:n+m})\mathrm{d}y_{n+1:n+m}. \qquad (2)$$

Ensuring that both of these conditions hold, allows us to invoke the Kolmogorov Extension and de-Finneti's theorems [25], hence prove that the model we defined is an exchangeable stochastic process. In this way we can guarantee that there is an underlying Bayesian model with an implied prior over global latent parameters $p_\theta(\mathbf{w})$ such that we can express the joint distribution in a conditional i.i.d. fashion, i.e. $p_\theta(y_1, \ldots, y_N|\mathbf{x}_1, \ldots, \mathbf{x}_N) = \int p_\theta(\mathbf{w}) \prod_{i=1}^N p(y_i|\mathbf{x}_i, \mathbf{w})\mathrm{d}\mathbf{w}$.

This constitutes the main objective of this work; how can we parametrize and optimize such distributions? Essentially, our target is to introduce dependence among the points of $\mathcal{D}$ in a manner that respects the two aforementioned conditions. We can then encode prior assumptions and inductive biases to the model by considering the relations among said points, a task much simpler than specifying a prior over latent global parameters $p_\theta(\mathbf{w})$. To this end, we introduce in the following our main contribution, the *Functional Neural Process* (FNP).

## 2.1 Designing the Functional Neural Process

On a high level the FNP follows the construction of a stochastic process as described at [11]; it posits a distribution over functions $h \in \mathcal{H}$ from $\mathbf{x}$ to $y$ by first selecting a "reference" set of points from $\mathcal{X}$, and then basing the probability distribution over $h$ around those points. This concept is similar to the "inducing inputs" that are used in sparse GPs [46, 51]. More specifically, let $R = \{\mathbf{x}_1^r, \ldots, \mathbf{x}_K^r\}$ be such a reference set and let $O = \mathcal{X} \setminus R$ be the "other" set, i.e. the set of all possible points that are not in $R$. Now let $\mathcal{D}_x = \{\mathbf{x}_1, \ldots, \mathbf{x}_N\}$ be any finite random set from $\mathcal{X}$, that constitutes our observed inputs. To facilitate the exposition we also introduce two more sets; $M = \mathcal{D}_x \setminus R$ that contains the points of $\mathcal{D}_x$ that are from $O$ and $B = R \cup M$ that contains all of the points in $\mathcal{D}_x$ and $R$. We provide a Venn diagram in Fig. 1. In the following we describe the construction of the model, shown in Fig. 2, and then prove that it corresponds to an infinitely exchangeable stochastic process.

**Embedding the inputs to a latent space** The first step of the FNP is to embed each of the $\mathbf{x}_i$ of $B$ independently to a latent representation $\mathbf{u}_i$

$$p_\theta(\mathbf{U}_B|\mathbf{X}_B) = \prod_{i \in B} p_\theta(\mathbf{u}_i|\mathbf{x}_i), \qquad (3)$$

where $p_\theta(\mathbf{u}_i|\mathbf{x}_i)$ can be any distribution, e.g. a Gaussian or a delta peak, where its parameters, e.g. the mean and variance, are given by a function of $\mathbf{x}_i$. This function can be any function, provided that it is flexible enough to provide a meaningful representation for $\mathbf{x}_i$. For this reason, we employ neural networks, as their representational capacity has been demonstrated on a variety of complex high dimensional tasks, such as natural image generation and classification.

**Constructing a graph of dependencies in the embedding space** The next step is to construct a dependency graph among the points in $B$; it encodes the correlations among the points in $\mathcal{D}$ that arise in the stochastic process. For example, in GPs such a correlation structure is encoded in the covariance matrix according to a kernel function $g(\cdot, \cdot)$ that measures the similarity between two inputs. In the FNP we adopt a different approach. Given the latent embeddings $\mathbf{U}_B$ that we obtained in the previous step we construct two directed graphs of dependencies among the points in $B$; a directed acyclic graph (DAG) $\mathbf{G}$ among the points in $R$ and a bipartite graph $\mathbf{A}$ from $R$ to $M$. These graphs are represented as random binary adjacency matrices, where e.g. $\mathbf{A}_{ij} = 1$ corresponds to the vertex $j$ being a parent for the vertex $i$. The distribution of the bipartite graph can be defined as

$$p(\mathbf{A}|\mathbf{U}_R, \mathbf{U}_M) = \prod_{i \in M} \prod_{j \in R} \mathrm{Bern}\left(\mathbf{A}_{ij}|g(\mathbf{u}_i, \mathbf{u}_j)\right). \qquad (4)$$

where $g(\mathbf{u}_i, \mathbf{u}_j)$ provides the probability that a point $i \in M$ depends on a point $j$ in the reference set $R$. This graph construction reminisces graphon [39] models, with however two important distinctions. Firstly, the embedding of each node is a vector rather than a scalar and secondly, the prior distribution over $\mathbf{u}$ is conditioned on an initial vertex representation $\mathbf{x}$ rather than being the same for all vertices. We believe that the latter is an important aspect, as it is what allows us to maintain enough information about the vertices and construct more informative graphs.

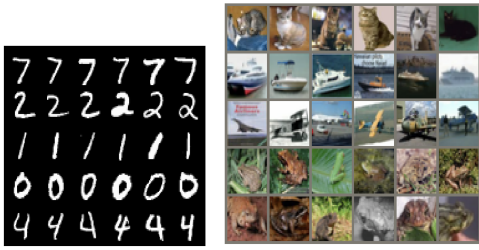 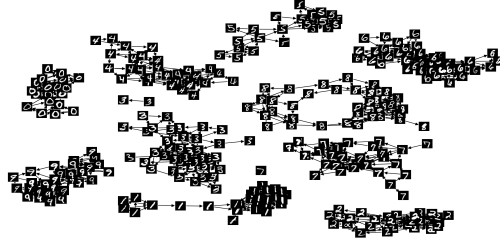

Figure 3: An example of the bipartite graph $\mathbf{A}$ that the FNP learns. The first column of each image is a query point and the rest are the five most probable parents from the $R$. We can see that the FNP associates same class inputs.

Figure 4: A DAG over $R$ on MNIST, obtained after propagating the means of $\mathbf{U}$ and thresholding edges that have less than $0.5$ probability in $\mathbf{G}$. We can see that FNP learns a meaningful $\mathbf{G}$ by connecting points that have the same class.

The DAG among the points in $R$ is a bit trickier, as we have to adopt a topological ordering of the vectors in $\mathbf{U}_R$ in order to avoid cycles. Inspired by the concept of stochastic orderings [43], we define an ordering according to a parameter free scalar projection $t(\cdot)$ of $\mathbf{u}$, i.e. $\mathbf{u}_i > \mathbf{u}_j$ when $t(\mathbf{u}_i) > t(\mathbf{u}_j)$. The function $t(\cdot)$ is defined as $t(\mathbf{u}_i) = \sum_k t_k(\mathbf{u}_{ik})$ where each individual $t_k(\cdot)$ is a monotonic function (e.g. the log CDF of a standard normal distribution); in this case we can guarantee that $\mathbf{u}_i > \mathbf{u}_j$ when individually for all of the dimensions $k$ we have that $\mathbf{u}_{ik} > \mathbf{u}_{jk}$ under $t_k(\cdot)$. This ordering can then be used in

$$p(\mathbf{G}|\mathbf{U}_R) = \prod_{i \in R} \prod_{j \in R, j \neq i} \mathrm{Bern}\left(\mathbf{G}_{ij}|\mathbb{I}[t(\mathbf{u}_i) > t(\mathbf{u}_j)]g(\mathbf{u}_i, \mathbf{u}_j)\right) \tag{5}$$

which leads into random adjacency matrices $\mathbf{G}$ that can be re-arranged into a triangular structure with zeros in the diagonal (i.e. DAGs). In a similar manner, such a DAG construction reminisces of digraphon models [6], a generalization of graphons to the directed case. The same two important distinctions still apply; we are using vector instead of scalar representations and the prior over the representation of each vertex $i$ depends on $\mathbf{x}_i$. It is now straightforward to bake in any relational inductive biases that we want our function to have by appropriately defining the $g(\cdot, \cdot)$ that is used for the construction of $\mathbf{G}$ and $\mathbf{A}$. For example, we can encode an inductive bias that neighboring points should be dependent by choosing $g(\mathbf{u}_i, \mathbf{u}_j) = \exp\left(-\frac{\tau}{2}\|\mathbf{u}_i - \mathbf{u}_j\|^2\right)$. This what we used in practice. We provide examples of the $\mathbf{A}, \mathbf{G}$ that FNPs learn in Figures 3, 4 respectively.

**Parametrizing the predictive distribution**    Having obtained the dependency graphs $\mathbf{A}, \mathbf{G}$, we are now interested in how to construct a predictive model that induces them. To this end, we parametrize predictive distributions for each target variable $y_i$ that explicitly depend on the reference set $R$ according to the structure of $\mathbf{G}$ and $\mathbf{A}$. This is realized via a local latent variable $\mathbf{z}_i$ that summarizes the context from the selected parent points in $R$ and their targets $\mathbf{y}_R$

$$\int p_\theta(\mathbf{y}_B, \mathbf{Z}_B|R, \mathbf{G}, \mathbf{A})\mathrm{d}\mathbf{Z}_B = \int p_\theta(\mathbf{y}_R, \mathbf{Z}_R|R, \mathbf{G})\mathrm{d}\mathbf{Z}_R \int p_\theta(\mathbf{y}_M, \mathbf{Z}_M|R, \mathbf{y}_R, \mathbf{A})\mathrm{d}\mathbf{Z}_M$$

$$= \prod_{i \in R} \int p_\theta\left(\mathbf{z}_i|\mathrm{par}_{\mathbf{G}_i}(R, \mathbf{y}_R)\right) p_\theta(y_i|\mathbf{z}_i)\mathrm{d}\mathbf{z}_i \prod_{j \in M} \int p_\theta\left(\mathbf{z}_j|\mathrm{par}_{\mathbf{A}_j}(R, \mathbf{y}_R)\right) p_\theta(y_j|\mathbf{z}_j)\mathrm{d}\mathbf{z}_j \tag{6}$$

where $\mathrm{par}_{\mathbf{G}_i}(\cdot), \mathrm{par}_{\mathbf{A}_j}(\cdot)$ are functions that return the parents of the point $i$, $j$ according to $\mathbf{G}, \mathbf{A}$ respectively. Notice that we are guaranteed that the decomposition to the conditionals at Eq. 6 is valid, since the DAG $\mathbf{G}$ coupled with $\mathbf{A}$ correspond to another DAG. Since permutation invariance in the parents is necessary for an overall exchangeable model, we define each distribution over $\mathbf{z}$, e.g. $p\left(\mathbf{z}_i|\mathrm{par}_{\mathbf{A}_i}(R, \mathbf{y}_R)\right)$, as an independent Gaussian distribution per dimension $k$ of $\mathbf{z}$[1]

$$p_\theta\left(\mathbf{z}_{ik}|\mathrm{par}_{\mathbf{A}_i}(R, \mathbf{y}_R)\right) = \mathcal{N}\left(\mathbf{z}_{ik}\middle| C_i \sum_{j \in R} \mathbf{A}_{ij}\mu_\theta(\mathbf{x}_j^r, y_j^r)_k, \exp\left(C_i \sum_{j \in R} \mathbf{A}_{ij}\nu_\theta(\mathbf{x}_j^r, y_j^r)_k\right)\right) \tag{7}$$

where the $\mu_\theta(\cdot, \cdot)$ and $\nu_\theta(\cdot, \cdot)$ are vector valued functions with a codomain in $\mathbb{R}^{|z|}$ that transform the data tuples of $R, \mathbf{y}_R$. The $C_i$ is a normalization constant with $C_i = (\sum_j A_{ij} + \epsilon)^{-1}$, i.e. it corresponds to the reciprocal of the number of parents of point $i$, with an extra small $\epsilon$ to avoid division by zero when a point has no parents. By observing Eq. 6 we can see that the prediction for a given $y_i$ depends on the input covariates $\mathbf{x}_i$ only indirectly via the graphs $\mathbf{G}, \mathbf{A}$ which are a function of $\mathbf{u}_i$. Intuitively, it encodes the inductive bias that predictions on points that are "far away", i.e. have very small probability of being connected to the reference set via $\mathbf{A}$, will default to an uninformative standard normal prior over $\mathbf{z}_i$ hence a constant prediction for $y_i$. This is similar to the behaviour that GPs with RBF kernels exhibit.

Nevertheless, Eq. 6 can also hinder extrapolation, something that neural networks can do well. In case extrapolation is important, we can always add a direct path by conditioning the prediction on $\mathbf{u}_i$, the latent embedding of $\mathbf{x}_i$, i.e. $p(y_i|\mathbf{z}_i, \mathbf{u}_i)$. This can serve as a middle ground where we can allow some extrapolation via $\mathbf{u}$. In general, it provides a knob, as we can now interpolate between GP and neural network behaviours by e.g. changing the dimensionalities of $\mathbf{z}$ and $\mathbf{u}$.

**Putting everything together: the FNP and FNP$^+$ models**   Now by putting everything together we arrive at the overall definitions of the two FNP models that we propose

$$\mathcal{FNP}_\theta(\mathcal{D}) := \sum_{\mathbf{G},\mathbf{A}} \int p_\theta(\mathbf{U}_B|\mathbf{X}_B)p(\mathbf{G},\mathbf{A}|\mathbf{U}_B)p_\theta(\mathbf{y}_B,\mathbf{Z}_B|R,\mathbf{G},\mathbf{A})\mathrm{d}\mathbf{U}_B\mathrm{d}\mathbf{Z}_B\mathrm{d}y_{i\in R\setminus\mathcal{D}_x}, \quad (8)$$

$$\mathcal{FNP}_\theta^+(\mathcal{D}) := \sum_{\mathbf{G},\mathbf{A}} \int p_\theta(\mathbf{U}_B,\mathbf{G},\mathbf{A}|\mathbf{X}_B)p_\theta(\mathbf{y}_B,\mathbf{Z}_B|R,\mathbf{U}_B,\mathbf{G},\mathbf{A})\mathrm{d}\mathbf{U}_B\mathrm{d}\mathbf{Z}_B\mathrm{d}y_{i\in R\setminus\mathcal{D}_x}, \quad (9)$$

where the first makes predictions according to Eq. 6 and the second further conditions on $\mathbf{u}$. Notice that besides the marginalizations over the latent variables and graphs, we also marginalize over any of the points in the reference set that are not part of the observed dataset $\mathcal{D}$. This is necessary for the proof of consistency that we provide later. For this work, we always chose the reference set to be a part of the dataset $\mathcal{D}$ so the extra integration is omitted. In general, the marginalization can provide a mechanism to include unlabelled data to the model which could be used to e.g. learn a better embedding $\mathbf{u}$ or "impute" the missing labels. We leave the exploration of such an avenue for future work. Having defined the models at Eq. 8, 9 we now prove that they both define valid permutation invariant stochastic processes by borrowing the methodology described at [11].

**Proposition 1.** *The distributions defined at Eq. 8, 9 are valid permutation invariant stochastic processes, hence they correspond to Bayesian models.*

*Proof sketch.*  The full proof can be found in the Appendix. Permutation invariance can be proved by noting that each of the terms in the products are permutation equivariant w.r.t. permutations of $\mathcal{D}$ hence each of the individual distributions defined at Eq. 8, 9 are permutation invariant due to the products. To prove consistency we have to consider two cases [11], the case where we add a point that is part of $R$ and the case where we add one that is not part of $R$. In the first case, marginalizing out that point will lead to the same distribution (as we were marginalizing over that point already), whereas in the second case the point that we are adding is a leaf in the dependency graph, hence marginalizing it doesn't affect the other points. $\qquad \square$

## 2.2   The FNPs in practice: fitting and predictions

Having defined the two models, we are now interested in how we can fit their parameters $\theta$ when we are presented with a dataset $\mathcal{D}$, as well as how to make predictions for novel inputs $\mathbf{x}^*$. For simplicity, we assume that $R \subseteq \mathcal{D}_x$ and focus on the FNP as the derivations for the FNP$^+$ are analogous. Notice that in this case we have that $B = \mathcal{D}_x = \mathbf{X}_\mathcal{D}$.

**Fitting the model to data**   Fitting the model parameters with maximum marginal likelihood is difficult, as the necessary integrals / sums of Eq.8 are intractable. For this reason, we employ variational inference and maximize the following lower bound to the marginal likelihood of $\mathcal{D}$

$$\mathcal{L} = \mathbb{E}_{q_\phi(\mathbf{U}_\mathcal{D},\mathbf{G},\mathbf{A},\mathbf{Z}_\mathcal{D}|\mathbf{X}_\mathcal{D})}[\log p_\theta(\mathbf{U}_\mathcal{D},\mathbf{G},\mathbf{A},\mathbf{Z}_\mathcal{D},\mathbf{y}_\mathcal{D}|\mathbf{X}_\mathcal{D}) - \log q_\phi(\mathbf{U}_\mathcal{D},\mathbf{G},\mathbf{A},\mathbf{Z}_\mathcal{D}|\mathbf{X}_\mathcal{D})], \quad (10)$$

with respect to the model parameters $\theta$ and variational parameters $\phi$. For a tractable lower bound, we assume that the variational posterior distribution $q_\phi(\mathbf{U}_\mathcal{D}, \mathbf{G}, \mathbf{A}, \mathbf{Z}_\mathcal{D}|\mathbf{X}_\mathcal{D})$ factorizes as $p_\theta(\mathbf{U}_\mathcal{D}|\mathbf{X}_\mathcal{D})p(\mathbf{G}|\mathbf{U}_R)p(\mathbf{A}|\mathbf{U}_\mathcal{D})q_\phi(\mathbf{Z}_\mathcal{D}|\mathbf{X}_\mathcal{D})$ with $q_\phi(\mathbf{Z}_\mathcal{D}|\mathbf{X}_\mathcal{D}) = \prod_{i=1}^{|\mathcal{D}|} q_\phi(\mathbf{z}_i|\mathbf{x}_i)$. This leads to

$$
\mathcal{L}_R + \mathcal{L}_{M|R} = \mathbb{E}_{p_\theta(\mathbf{U}_R, \mathbf{G}|\mathbf{X}_R)q_\phi(\mathbf{Z}_R|\mathbf{X}_R)}[\log p_\theta(\mathbf{y}_R, \mathbf{Z}_R|R, \mathbf{G}) - \log q_\phi(\mathbf{Z}_R|\mathbf{X}_R)] + \tag{11}
$$
$$
+ \mathbb{E}_{p_\theta(\mathbf{U}_\mathcal{D}, \mathbf{A}|\mathbf{X}_\mathcal{D})q_\phi(\mathbf{Z}_M|\mathbf{X}_M)}[\log p_\theta(\mathbf{y}_M|\mathbf{Z}_M) + \log p_\theta(\mathbf{Z}_M|\mathrm{par}_\mathbf{A}(R, \mathbf{y}_R)) - \log q_\phi(\mathbf{Z}_M|\mathbf{X}_M)]
$$

where we decomposed the lower bound into the terms for the reference set $R$, $\mathcal{L}_R$, and the terms that correspond to $M$, $\mathcal{L}_{M|R}$. For large datasets $\mathcal{D}$ we are interested in doing efficient optimization of this bound. While the first term is not, in general, amenable to minibatching, the second term is. As a result, we can use minibatches that scale according to the size of the reference set $R$. We provide more details in the Appendix.

In practice, for all of the distributions over $\mathbf{u}$ and $\mathbf{z}$, we use diagonal Gaussians, whereas for $\mathbf{G}, \mathbf{A}$ we use the concrete / Gumbel-softmax relaxations [34, 21] during training. In this way we can jointly optimize $\theta, \phi$ with gradient based optimization by employing the pathwise derivatives obtained with the reparametrization trick [24, 42]. Furthermore, we tie most of the parameters $\theta$ of the model and $\phi$ of the inference network, as the regularizing nature of the lower bound can alleviate potential overfitting of the model parameters $\theta$. More specifically, for $p_\theta(\mathbf{u}_i|\mathbf{x}_i)$, $q_\phi(\mathbf{z}_i|\mathbf{x}_i)$ we share a neural network torso and have two output heads, one for each distribution. We also parametrize the priors over the latent $\mathbf{z}$ in terms of the $q_\phi(\mathbf{z}_i|\mathbf{x}_i)$ for the points in $R$; the $\mu_\theta(\mathbf{x}_i^r, y_i^r), \nu_\theta(\mathbf{x}_i^r, y_i^r)$ are both defined as $\mu_q(\mathbf{x}_i^r) + \mu_y^r, \nu_q(\mathbf{x}_i^r) + \nu_y^r$, where $\mu_q(\cdot), \nu_q(\cdot)$ are the functions that provide the mean and variance for $q_\phi(\mathbf{z}_i|\mathbf{x}_i)$ and $\mu_y^r, \nu_y^r$ are linear embeddings of the labels.

It is interesting to see that the overall bound at Eq. 11 reminisces the bound of a latent variable model such as a variational autoencoder (VAE) [24, 42] or a deep variational information bottleneck model (VIB) [1]. We aim to predict the label $y_i$ of a given point $\mathbf{x}_i$ from its latent code $\mathbf{z}_i$ where the prior, instead of being globally the same as in [24, 42, 1], it is conditioned on the parents of that particular point. The conditioning is also intuitive, as it is what converts the i.i.d. to the more general exchangeable model. This is also similar to the VAE for unsupervised learning described at associative compression networks (ACN) [16] and reminisces works on few-shot learning [4].

**The posterior predictive distribution** In order to perform predictions for unseen points $\mathbf{x}^*$, we employ the posterior predictive distribution of FNPs. More specifically, we can show that by using Bayes rule, the predictive distribution of the FNPs has the following simple form

$$
\sum_{\mathbf{a}^*} \int p_\theta(\mathbf{U}_R, \mathbf{u}^*|\mathbf{X}_R, \mathbf{x}^*)p(\mathbf{a}^*|\mathbf{U}_R, \mathbf{u}^*)p_\theta(\mathbf{z}^*|\mathrm{par}_{\mathbf{a}^*}(R, \mathbf{y}_R))p_\theta(y^*|\mathbf{z}^*)\mathrm{d}\mathbf{U}_R\mathrm{d}\mathbf{u}^*\mathrm{d}\mathbf{z}^* \tag{12}
$$

where $\mathbf{u}$ are the representations given by the neural network and $\mathbf{a}^*$ is the binary vector that denotes which points from $R$ are the parents of the new point. We provide more details in the Appendix. Intuitively, we first project the reference set and the new point on the latent space $\mathbf{u}$ with a neural network and then make a prediction $y^*$ by basing it on the parents from $R$ according to $\mathbf{a}^*$. This predictive distribution reminisces the models employed in few-shot learning [53].

## 3 Related work

There has been a long line of research in Bayesian Neural Networks (BNNs) [15, 5, 23, 19, 31, 44]. A lot of works have focused on the hard task of posterior inference for BNNs, by positing more flexible posteriors [31, 44, 30, 56, 3]. The exploration of more involved priors has so far not gain much traction, with the exception of a handful of works [23, 29, 2, 17]. For flexible stochastic processes, we have a line of works that focus on (scalable) Gaussian Processes (GPs); these revolve around sparse GPs [46, 51], using neural networks to parametrize the kernel of a GP [55, 54], employing finite rank approximations to the kernel [9, 18] or parametrizing kernels over structured data [35, 52]. Compared to such approaches, FNPs can in general be more scalable due to not having to invert a matrix for prediction and, furthermore, they can easily support arbitrary likelihood models (e.g. for discrete data) without having to consider appropriate transformations of a base Gaussian distribution (which usually requires further approximations).

There have been interesting recent works that attempt to merge stochastic processes and neural networks. Neural Processes (NPs) [14] define distributions over global latent variables in terms

of subsets of the data, while Attentive NPs [22] extend NPs with a deterministic path that has a cross-attention mechanism among the datapoints. In a sense, FNPs can be seen as a variant where we discard the global latent variables and instead incorporate cross-attention in the form of a dependency graph among local latent variables. Another line of works is the Variational Implicit Processes (VIPs) [32], which consider BNN priors and then use GPs for inference, and functional variational BNNs (fBNNs) [47], which employ GP priors and use BNNs for inference. Both methods have their drawbacks, as with VIPs we have to posit a meaningful prior over global parameters and the objective of fBNNs does not always correspond to a bound of the marginal likelihood. Finally, there is also an interesting line of works that study wide neural networks with random Gaussian parameters and discuss their equivalences with Gaussian Processes [38, 27], as well as the resulting kernel [20].

Similarities can be also seen at other works; Associative Compression Networks (ACNs) [16] employ similar ideas for generative modelling with VAEs and conditions the prior over the latent variable of a point to its nearest neighbors. Correlated VAEs [50] similarly employ a (a-priori known) dependency structure across the latent variables of the points in the dataset. In few-shot learning, metric-based approaches [53, 4, 48, 45, 26] similarly rely on similarities w.r.t. a reference set for predictions.

## 4 Experiments

We performed two main experiments in order to verify the effectiveness of FNPs. We implemented and compared against 4 baselines: a standard neural network (denoted as NN), a neural network trained and evaluated with Monte Carlo (MC) dropout [13] and a Neural Process (NP) [14] architecture. The architecture of the NP was designed in a way that is similar to the FNP. For the first experiment we explored the inductive biases we can encode in FNPs by visualizing the predictive distributions in toy 1d regression tasks. For the second, we measured the prediction performance and uncertainty quality that FNPs can offer on the benchmark image classification tasks of MNIST and CIFAR 10. For this experiment, we also implemented and compared against a Bayesian neural network trained with variational inference [5]. We provide the experimental details in the Appendix.

For all of the experiments in the paper, the NP was trained in a way that mimics the FNP, albeit we used a different set $R$ at every training iteration in order to conform to the standard NP training regime. More specifically, a random amount from 3 to $num(R)$ points were selected as a context from each batch, with $num(R)$ being the maximum amount of points allocated for $R$. For the toy regression task we set $num(R) = N - 1$.

**Exploring the inductive biases in toy regression** To visually access the inductive biases we encode in the FNP we experiment with two toy 1-d regression tasks described at [40] and [19] respectively. The generative process of the first corresponds to drawing 12 points from $U[0, 0.6]$, 8 points from $U[0.8, 1]$ and then parametrizing the target as $y_i = x_i + \epsilon + \sin(4(x_i + \epsilon)) + \sin(13(x_i + \epsilon))$ with $\epsilon \sim \mathcal{N}(0, 0.03^2)$. This generates a nonlinear function with "gaps" in between the data where we, ideally, want the uncertainty to be high. For the second we sampled 20 points from $U[-4, 4]$ and then parametrized the target as $y_i = x_i^3 + \epsilon$, where $\epsilon \sim \mathcal{N}(0, 9)$. For all of the models we used a heteroscedastic noise model. Furthermore, due to the toy nature of this experiment, we also included a Gaussian Process (GP) with an RBF kernel. We used 50 dimensions for the global latent of NP for the first task and 10 dimensions for the second. For the FNP models we used $3, 50$ dimensions for the $\mathbf{u}, \mathbf{z}$ for the first task and $3, 10$ for the second. For the reference set $R$ we used 10 random points for the FNPs and the full dataset for the NP.

The results we obtain are presented in Figure 5. We can see that for the first task the FNP with the RBF function for $g(\cdot, \cdot)$ has a behaviour that is very similar to the GP. We can also see that in the second task it has the tendency to quickly move towards a flat prediction outside the areas where we observe points, something which we argued about at Section 2.1. This is not the case for MC-dropout or NP where we see a more linear behaviour on the uncertainty and erroneous overconfidence, in the case of the first task, in the areas in-between the data. Nevertheless, they do seem to extrapolate better compared to the FNP and GP. The FNP$^+$ seems to combine the best of both worlds as it allows for extrapolation and GP like uncertainty, although a free bits [7] modification of the bound for $\mathbf{z}$ was helpful in encouraging the model to rely more on these particular latent variables. Empirically, we observed that adding more capacity on $\mathbf{u}$ can move the FNP$^+$ closer to the behaviour we observe for MC-dropout and NPs. In addition, increasing the amount of model parameters $\theta$ can make FNPs overfit, a fact that can result into a reduction of predictive uncertainty.

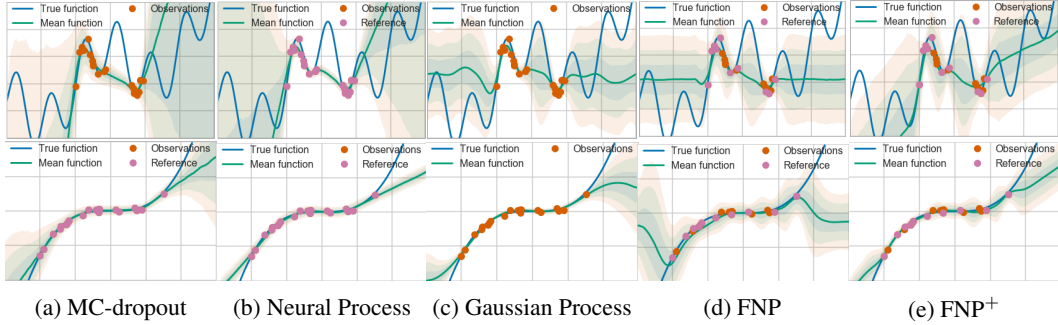

|  (a) MC-dropout | (b) Neural Process | (c) Gaussian Process | (d) FNP | (e) FNP$^+$ |

Figure 5: Predictive distributions for the two toy regression tasks according to the different models we considered. Shaded areas correspond to $\pm\,3$ standard deviations.

**Prediction performance and uncertainty quality**  For the second task we considered the image classification of MNIST and CIFAR 10. For MNIST we used a LeNet-5 architecture that had two convolutional and two fully connected layers, whereas for CIFAR we used a VGG-like architecture that had 6 convolutional and two fully connected. In both experiments we used 300 random points from $\mathcal{D}$ as $R$ for the FNPs and for NPs, in order to be comparable, we randomly selected up to 300 points from the current batch for the context points during training and used the same 300 points as FNPs for evaluation. The dimensionality of $\mathbf{u}, \mathbf{z}$ was $32, 64$ for the FNP models in both datasets, whereas for the NP the dimensionality of the global variable was $32$ for MNIST and $64$ for CIFAR.

As a proxy for the uncertainty quality we used the task of out of distribution (o.o.d.) detection; given the fact that FNPs are Bayesian models we would expect that their epistemic uncertainty will increase in areas where we have no data (i.e. o.o.d. datasets). The metric that we report is the average entropy on those datasets as well as the area under an ROC curve (AUCR) that determines whether a point is in or out of distribution according to the predictive entropy. Notice that it is simple to increase the first metric by just learning a trivial model but that would be detrimental for AUCR; in order to have good AUCR the model must have low entropy on the in-distribution test set but high entropy on the o.o.d. datasets. For the MNIST model we considered notMNIST, Fashion MNIST, Omniglot, Gaussian $\mathcal{N}(0, 1)$ and uniform $U[0, 1]$ noise as o.o.d. datasets whereas for CIFAR 10 we considered SVHN, a tinyImagenet resized to 32 pixels, iSUN and similarly Gaussian and uniform noise. The summary of the results can be seen at Table 1.

Table 1: Accuracy and uncertainty on MNIST and CIFAR 10 from 100 posterior predictive samples. For the all of the datasets the first column is the average predictive entropy whereas for the o.o.d. datasets the second is the AUCR and for the in-distribution it is the test error in %.

|  | NN | MC-Dropout | VI BNN | NP | FNP | FNP$^+$ |
|---|---|---|---|---|---|---|
| MNIST | 0.01 / 0.6 | **0.05 / 0.5** | 0.02 / 0.6 | 0.01 / 0.6 | 0.04 / 0.7 | 0.02 / 0.7 |
| nMNIST | 1.03 / 99.73 | 1.30 / 99.48 | 1.33 / 99.80 | 1.31 / 99.90 | 1.94 / 99.90 | **1.77 / 99.96** |
| fMNIST | 0.81 / 99.16 | 1.23 / 99.07 | 0.92 / 98.61 | 0.71 / 98.98 | **1.85 / 99.66** | 1.55 / 99.58 |
| Omniglot | 0.71 / 99.44 | 1.18 / 99.29 | 1.61 / 99.91 | 0.86 / 99.69 | 1.87 / 99.79 | **1.71 / 99.92** |
| Gaussian | 0.99 / 99.63 | **2.03 / 100.0** | **1.77 / 100.0** | 1.58 / 99.94 | 1.94 / 99.86 | **2.03 / 100.0** |
| Uniform | 0.85 / 99.65 | 0.65 / 97.58 | 1.41 / 99.87 | 1.46 / 99.96 | 2.11 / 99.98 | **1.88 / 99.99** |
| Average | 0.9±0.1 / 99.5±0.1 | 1.3±0.2 / 99.1±0.4 | 1.4±0.1 / 99.6±0.3 | 1.2±0.2 / 99.7±0.2 | 1.9±0.1 / 99.8±0.1 | **1.8±0.1 / 99.9±0.1** |
| CIFAR10 | 0.05 / 6.9 | 0.06 / 7.0 | **0.06 / 6.4** | 0.06 / 7.5 | 0.18 / 7.2 | 0.08 / 7.2 |
| SVHN | 0.44 / 93.1 | 0.42 / 91.3 | 0.45 / 91.8 | 0.38 / 90.2 | **1.09 / 94.3** | 0.42 / 89.8 |
| tImag32 | 0.51 / 92.7 | 0.59 / 93.1 | 0.52 / 91.9 | 0.45 / 89.8 | **1.20 / 94.0** | 0.74 / 93.8 |
| iSUN | 0.52 / 93.2 | 0.59 / 93.1 | 0.57 / 93.2 | 0.47 / 90.8 | **1.30 / 95.1** | 0.81 / 94.8 |
| Gaussian | 0.01 / 72.3 | 0.05 / 72.1 | 0.76 / 96.9 | 0.37 / 91.9 | 1.13 / 95.4 | **0.96 / 97.9** |
| Uniform | **0.93 / 98.4** | 0.08 / 77.3 | 0.65 / 96.1 | 0.17 / 87.8 | 0.71 / 89.7 | **0.99 / 98.4** |
| Average | 0.5±0.2 / 89.9±4.5 | 0.4±0.1 / 85.4±4.5 | 0.6±0.1 / 94±1.1 | 0.4±0.1 / 90.1±0.7 | 1.1±0.1 / 93.7±1.0 | **0.8±0.1 / 94.9±1.6** |

We observe that both FNPs have comparable accuracy to the baseline models while having higher average entropies and AUCR on the o.o.d. datasets. FNP$^+$ in general seems to perform better than FNP. The FNP did have a relatively high in-distribution entropy for CIFAR 10, perhaps denoting that a larger $R$ might be more appropriate. We further see that the FNPs have almost always better AUCR than all of the baselines we considered. Interestingly, out of all the non-noise o.o.d. datasets

we did observe that Fashion MNIST and SVHN, were the hardest to distinguish on average across all the models. This effect seems to agree with the observations from [36], although more investigation is required. We also observed that, sometimes, the noise datasets on all of the baselines can act as "adversarial examples" [49] thus leading to lower entropy than the in-distribution test set (e.g. Gaussian noise for the NN on CIFAR 10). FNPs did have a similar effect on CIFAR 10, e.g. the FNP on uniform noise, although to a much lesser extent. We leave the exploration of this phenomenon for future work. It should be mentioned that other advances in o.o.d. detection, e.g. [28, 8], are orthogonal to FNPs and could further improve performance.

We further performed additional experiments in order to better disentangle the performance differences between NPs and FNPs: we trained an NP with the same fixed reference set $R$ as the FNPs throughout training, as well as an FNP$^+$ where we randomly sample a new $R$ for every batch (akin to the NP) and use the same $R$ as the NP for evaluation. While we argued in the construction of the FNPs that with a fixed $R$ we can obtain a stochastic process, we could view the case with random $R$ as an ensemble of stochastic processes, one for each realization of $R$. The results from these models can be seen at Table 2. On the one hand, the FNP$^+$ still provides robust uncertainty while the randomness in $R$ seems to improve the o.o.d. detection, possibly due to the implicit regularization. On the other hand the fixed $R$ seems to hurt the NP, as the o.o.d. detection decreased, similarly hinting that the random $R$ has beneficial regularizing effects.

Finally, we provide some additional insights after doing ablation studies on MNIST w.r.t. the sensitivity to the number of points in $R$ for NP, FNP and FNP$^+$, as well as

Table 2: Results obtained by training a NP model with a fixed reference set (akin to FNP) and a FNP$^+$ model with a random reference set (akin to NP).

|          | NP fixed $R$ | FNP$^+$ random $R$ |
|----------|--------------|--------------------|
| MNIST    | 0.01 / 0.6   | 0.02 / 0.8         |
| nMNIST   | 1.09 / 99.78 | 2.20 / 100.0       |
| fMNIST   | 0.64 / 98.34 | 1.58 / 99.78       |
| Omniglot | 0.79 / 99.53 | 2.06 / 99.99       |
| Gaussian | 1.79 / 99.96 | 2.28 / 100.0       |
| Uniform  | 1.42 / 99.93 | 2.23 / 100.0       |
| CIFAR10  | 0.07 / 7.5   | 0.09 / 6.9         |
| SVHN     | 0.46 / 91.5  | 0.56 / 91.4        |
| tImag32  | 0.55 / 91.5  | 0.77 / 93.4        |
| iSUN     | 0.60 / 92.6  | 0.83 / 94.0        |
| Gaussian | 0.20 / 87.2  | 1.23 / 99.1        |
| Uniform  | 0.53 / 94.3  | 0.90 / 97.2        |

varying the amount of dimensions for $\mathbf{u}, \mathbf{z}$ in the FNP$^+$. The results can be found in the Appendix. We generally observed that NP models have lower average entropy at the o.o.d. datasets than both FNP and FNP$^+$ irrespective of the size of $R$. The choice of $R$ seems to be more important for the FNPs rather than NPs, with FNP needing a larger $R$, compared to FNP$^+$, to fit the data well. In general, it seems that it is not the quantity of points that matters but rather the quality; the performance did not always increase with more points. This supports the idea of a "coreset" of points, thus exploring ideas to infer it is a promising research direction that could improve scalability and alleviate the dependence of FNPs on a reasonable $R$. As for the trade-off between $\mathbf{z}, \mathbf{u}$ in FNP$^+$; a larger capacity for $\mathbf{z}$, compared to $\mathbf{u}$, leads to better uncertainty whereas the other way around seems to improve accuracy. These observations are conditioned on having a reasonably large $\mathbf{u}$, which facilitates for meaningful $\mathbf{G}, \mathbf{A}$.

## 5    Discussion

We presented a novel family of exchangeable stochastic processes, the Functional Neural Processes (FNPs). In contrast to NPs [14] that employ global latent variables, FNPs operate by employing local latent variables along with a dependency structure among them, a fact that allows for easier encoding of inductive biases. We verified the potential of FNPs experimentally, and showed that they can serve as competitive alternatives. We believe that FNPs open the door to plenty of exciting avenues for future research; designing better function priors by e.g. imposing a manifold structure on the FNP latents [12], extending FNPs to unsupervised learning by e.g. adapting ACNs [16] or considering hierarchical models similar to deep GPs [10].

**Acknowledgments**

We would like to thank Patrick Forré for helpful discussions over the course of this project and Peter Orbanz, Benjamin Bloem-Reddy for helpful discussions during a preliminary version of this work. We would also like to thank Daniel Worrall, Tim Bakker and Stephan Alaniz for helpful feedback on an initial draft.

## Footnotes

[1]The factorized Gaussian distribution was chosen for simplicity, and it is not a limitation. Any distribution is valid for $\mathbf{z}$ provided that it defines a permutation invariant probability density w.r.t. the parents.

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
