[Supplementary Material · The_Functional_Neural_Process_appendix.pdf]

# The Functional Neural Process

**Christos Louizos**
University of Amsterdam
TNO Intelligent Imaging
c.louizos@uva.nl

**Xiahan Shi**
Bosch Center for Artificial Intelligence
UvA-Bosch Delta Lab
xiahan.shi@de.bosch.com

**Klamer Schutte**
TNO Intelligent Imaging
klamer.schutter@tno.nl

**Max Welling**
University of Amsterdam
Qualcomm
m.welling@uva.nl

# Appendix

## A  Experimental details

Throughout the experiments, the architectures for the FNP and FNP$^+$ were constructed as follows. We used a neural network torso in order to obtain an intermediate hidden representation $\mathbf{h}$ of the inputs $\mathbf{x}$ and then parametrized two linear output layers, one that lead to the parameters of $p(\mathbf{u}|\mathbf{x})$ and one that lead to the parameters of $q(\mathbf{z}|\mathbf{x})$, both of which were fully factorized Gaussians. The function $g(\cdot, \cdot)$ for the Bernoulli probabilities was set to an RBF, i.e. $g(\mathbf{u}_i, \mathbf{u}_j) = \exp(-.5\tau\|\mathbf{u}_i - \mathbf{u}_j\|^2)$, where $\tau$ was optimized to maximize the lower bound. The temperature of the binary concrete / Gumbel-softmax relaxation was kept at $0.3$ throughout training and we used the log CDF of a standard normal as the $\tau_k(\cdot)$ for $\mathbf{G}$. For the classifiers $p(y|\mathbf{z}), p(y|\mathbf{z}, \mathbf{u})$ we used a linear model that operated on top of ReLU$(\mathbf{z})$ or ReLU$([\mathbf{z}, \mathbf{u}])$ respectively. We used a single Monte Carlo sample for each batch during training in order to estimate the bound of FNPs. We similarly used a single sample for the NP, MC-dropout and the variationally trained Bayesian neural network (VI BNN). All of the models were implemented in PyTorch and were run across six Titan X (Pascal) GPUs (one GPU per model).

The NN, MC-dropout and VI BNN had the same torso and classifier as the FNPs. As the NP has not been previously employed in the settings we considered, we designed the architecture in a way that is similar to the FNP. More specifically, we used the same neural network torso to provide an intermediate representation $\mathbf{h}$ for the inputs $\mathbf{x}$. To obtain the global embedding $\mathbf{r}$ we concatenated the labels $\mathbf{y}$ to obtain $\tilde{\mathbf{h}} = [\mathbf{h}, \mathbf{y}]$, projected $\tilde{\mathbf{h}}$ to $256$ dimensions with a linear layer and then computed the average of each dimension across the context. The parameters of the distribution over the global latent variables $\boldsymbol{\theta}$ were then given by a linear layer acting on top of ReLU$(\mathbf{r})$. After sampling $\boldsymbol{\theta}$ we then used a linear classifier that operated on top of $[\mathbf{h}, \text{ReLU}(\boldsymbol{\theta})]$.

In the regression experiment for the initial transformation of $x$ we used 100 ReLUs for both NP and FNP models via a single layer MLP, whereas for the regressor we used a linear layer for NP (more capacity lead to overfitting and a decrease in predictive uncertainty) and a single hidden layer MLP of 100 ReLUs for the FNPs. For the MC-dropout network used a single hidden layer MLP of 100 units and we applied dropout with a rate of 0.5 at the hidden layer. In all of the neural networks models, the heteroscedastic noise was parametrized according to $\sigma = .1 + 0.9 \log(1 + \exp(d))$, where $d$ was a neural network output. For the GP, we optimized the kernel lengthscale according to the marginal likelihood. We also found it beneficial to apply a soft-free bits [1] modification of the bound to help with the optimization of $\mathbf{z}$, where we allowed 1 free bit on average across all dimensions and batch elements for the FNP and 4 for the FNP$^+$.

For the MNIST experiment, the model architecture was a 20C5 - MP2 - 50C5 - MP2 - 500FC - Softmax, where 20C5 corresponds to a convolutional layer of 20 output feature maps with a kernel size of 5, MP2 corresponds to max pooling with a size of 2, 500FC corresponds to fully connected layer of 500 output units and Softmax corresponds to the output layer. The initial representation of $\mathbf{x}$ for the NP and FNPs was provided by the penultimate layer of the network. For the MC-dropout network we applied 0.5 dropout to every layer, whereas for the VI BNN we performed variational inference over all of the parameters. The number of points in $R$ was set to 300, a value that was determined from a range of $[50, 100, 200, 300, 500]$ by judging the performance of the NP and FNP models on the MNIST / notMNIST pair. For the FNP we used minibatches of 100 points from $M$, while we always appended the full $R$ to each of those batches. For the NP, since we were using a random set of contexts every time, we used a batch size of 400 points, where, in order to be comparable to the FNP, we randomly selected up to 300 points from the current batch for the context points during training and used the same 300 points as FNP for evaluation. We set the upper bound of training epochs for the FNPs, NN, MC-dropout and VI BNN networks to 100 epochs, and 200 epochs for the NP as it did less parameter updates per epoch than the others. Optimization was done with Adam [4] using the default hyperparameters. We further did early stopping according to the accuracy on the validation set and no other regularization was employed. Finally, we also employed a soft-free bits [1] modification of the bound to help with the optimization; for the $\mathbf{z}$ of FNPs we allowed 1 free bit on average across all dimensions and batch elements throughout training whereas for the VI BNN we allowed 1 free bit on average across all of the parameters.

The architecture for the CIFAR 10 experiment was a 2x(128C3) - MP2 - 2x(256C3) - MP2 - 2x(512C3) - MP2 - 1024FC - Softmax along with batch normalization [3] employed after every layer (besides the output one). Similarly to the MNIST experiment, the initial representation of $\mathbf{x}$ for the NP and FNPs was provided by the penultimate layer of each of the networks. We didn't optimize any hyperparameters for these experiments and used the same number of reference points, free bits, amount of epochs, regularization and early stopping criteria we used at MNIST. For the MC-dropout network we applied dropout with a rate of 0.2 at the beginning of each stack of convolutional layers that shared the same output channels and with a rate of 0.5 before every fully connected layer. For the VI BNN, we performed variational inference over the layers were dropout was originally employed in the MC-dropout network, and we performed maximum likelihood for the rest. Optimization was done with Adam with an initial learning rate of 0.001 that was decayed by a factor of 10 every thirty epochs for the NN, MC-Dropout, VI BNN and FNPs and every 60 epochs for the NP. We also performed data augmentation during training by doing random cropping with a padding of 4 pixels and random horizontal flips for both the reference and other points. We did not do any data augmentation during test time. The images were further normalized by subtracting the mean and by dividing with the standard deviation of each channel, computed across the training dataset.

# B  Ablation study on MNIST

In this section we provide the additional results we obtained on MNIST during the ablation study. The discussion of the results can be found in the main text. We measured the sensitivity of NPs and FNPs to the size of the reference set $R$, the trade-offs we obtain by varying the dimensionalities of $\mathbf{u}, \mathbf{z}$ for the FNP$^+$, and finally the deviation of the scores for the FNPs after 5 replications, where on each one we select a different subset of size 300 (the size we used for the experiments in the main paper) of the training set as $R$. The results can be seen at Table 1, Table 2 and Table 3.

Table 1: Test error and uncertainty quality as a function of the size of the reference set $R$. For the o.o.d. entropy and AUCR we report the mean and standard error across all of the o.o.d. datasets.

| (a) Error % | | | | (b) o.o.d. entropy | | | | (c) AUCR | | |
|---|---|---|---|---|---|---|---|---|---|---|
| # $R$ | NP | FNP | FNP$^+$ | # $R$ | NP | FNP | FNP$^+$ | # $R$ | NP | FNP | FNP$^+$ |
| 50 | 0.6 | 30.6 | 0.7 | 50 | 1.0±0.2 | 2.1±0.0 | 1.6±0.1 | 50 | 99.4±0.3 | 80.0±0.3 | 99.7±0.2 |
| 100 | 0.6 | 0.9 | 0.9 | 100 | 1.4±0.3 | 1.8±0.1 | 2.0±0.1 | 100 | 99.7±0.2 | 99.6±0.1 | 99.9±0.1 |
| 200 | 0.4 | 0.8 | 0.7 | 200 | 0.9±0.2 | 1.8±0.1 | 1.6±0.1 | 200 | 99.5±0.2 | 99.8±0.1 | 99.8±0.1 |
| 500 | 0.5 | 0.9 | 0.7 | 500 | 0.8±0.1 | 1.7±0.1 | 1.3±0.1 | 500 | 99.5±0.2 | 99.8±0.1 | 99.4±0.3 |

Table 2: Test error and uncertainty quality as a function of the size of $\mathbf{u}, \mathbf{z}$ for the FNP$^+$. We used the same $R$ as the one for the experiments in the main text.

| $\mathbf{u}, \mathbf{z}$ | Error % | o.o.d. entropy | AUCR |
|---|---|---|---|
| $32, 64$ | 0.7 | $1.8 \pm 0.1$ | $99.9 \pm 0.1$ |
| $64, 32$ | 0.7 | $1.2 \pm 0.3$ | $99.3 \pm 0.6$ |
| $16, 80$ | 0.7 | $2.0 \pm 0.1$ | $99.6 \pm 0.4$ |
| $8, 88$ | 0.8 | $1.1 \pm 0.0$ | $99.4 \pm 0.1$ |
| $2, 94$ | 4.7 | $0.4 \pm 0.0$ | $91.4 \pm 2.2$ |

Table 3: Average and standard error for the FNP models after 5 replications with different reference sets $R$ of size 300.

| | FNP | FNP$^+$ |
|---|---|---|
| MNIST | $0.02 \pm 0.0$ / $0.7 \pm 0.0$ | $0.02 \pm 0.0$ / $0.7 \pm 0.0$ |
| nMNIST | $1.95 \pm 0.06$ / $99.93 \pm 0.03$ | $1.97 \pm 0.05$ / $99.97 \pm 0.02$ |
| fMNIST | $1.69 \pm 0.05$ / $99.43 \pm 0.10$ | $1.63 \pm 0.04$ / $99.58 \pm 0.07$ |
| Omniglot | $1.88 \pm 0.04$ / $99.86 \pm 0.04$ | $1.85 \pm 0.06$ / $99.90 \pm 0.03$ |
| Gaussian | $1.95 \pm 0.14$ / $99.81 \pm 0.16$ | $2.07 \pm 0.02$ / $99.98 \pm 0.02$ |
| Uniform | $1.99 \pm 0.06$ / $99.96 \pm 0.02$ | $1.95 \pm 0.06$ / $99.96 \pm 0.02$ |
| CIFAR10 | $0.17 \pm 0.01$ / $7.5 \pm 0.08$ | $0.08 \pm 0.01$ / $7.3 \pm 0.04$ |
| SVHN | $0.86 \pm 0.05$ / $90.74 \pm 0.81$ | $0.51 \pm 0.04$ / $91.3 \pm 0.76$ |
| tImag32 | $1.22 \pm 0.02$ / $94.49 \pm 0.29$ | $0.69 \pm 0.02$ / $92.6 \pm 0.39$ |
| iSUN | $1.33 \pm 0.02$ / $95.71 \pm 0.24$ | $0.75 \pm 0.02$ / $93.8 \pm 0.38$ |
| Gaussian | $1.05 \pm 0.10$ / $93.73 \pm 1.29$ | $0.60 \pm 0.08$ / $93.6 \pm 1.09$ |
| Uniform | $0.85 \pm 0.16$ / $89.43 \pm 4.20$ | $0.61 \pm 0.11$ / $93.4 \pm 1.89$ |

## C  The Functional Neural Process is an exchangeable stochastic process

**Proposition.** *The distributions defined in Eq.8, 9 define valid permutation invariant stochastic processes, hence they correspond to Bayesian models.*

*Proof.* In order to prove the proposition we will rely on de Finetti's and Kolmogorov Extension Theorems [5] and show that $p(\mathbf{y}_{\mathcal{D}}|\mathbf{X}_{\mathcal{D}})$ is permutation invariant and its marginal distributions are consistent under marginalization. We will focus on FNP as the proof for FNP$^+$ is analogous. As a reminder, we previously defined $R$ to be a set of reference inputs $R = \{\mathbf{x}_1^r, \ldots, \mathbf{x}_K^r\}$, we defined $\mathcal{D}_x$ to be the set of observed inputs, and we also defined the auxiliary sets $M = \mathcal{D}_x \setminus R$, the set of all inputs in the observed dataset that are not a part of the reference set $R$, and $B = R \cup M$, the set of all points in the reference and observed dataset.

We will start with the permutation invariance. It will suffice to show that each of the individual probability densities described at Section 2.1 are permutation equivariant, as the products / sums will then make the overall probability permutation invariant. Without loss of generality we will assume that the elements in the set $B$ are arranged as $B = \{\mathcal{D}_x, R \setminus \mathcal{D}_x\}$. Consider applying a permutation $\sigma(\cdot)$ over $\mathcal{D}$, $\tilde{\mathcal{D}} = \sigma(\mathcal{D})$; this will also induce the same permutation over $\mathcal{D}_x$, hence we will have that $\sigma(B) = \{\sigma(\mathcal{D}_x), R \setminus \mathcal{D}_x\}$. Now consider the fact that in the FNP each individual $\mathbf{u}_i$ is a function, let it be $f(\cdot)$, of the values of $\mathbf{x}_i$; as a result we will have that:

$$f(\sigma(B)) = \sigma(f(B)),  \tag{1}$$

i.e. the latent variables $\mathbf{u}$ are permutation equivariant w.r.t. $B$. Continuing to the latent adjacency matrices $\mathbf{G}, \mathbf{A}$; in the FNP each particular element $i, j$ of these is a function of the values of the specific $\mathbf{u}_i, \mathbf{u}_j$. As a result, we will also have permutation equivariance for the rows / columns of $\mathbf{G}, \mathbf{A}$. Now since $\mathbf{G}, \mathbf{A}$ are essentially used as a way to factorize the joint distribution over the $\mathbf{z}_i$ in $B$ and given the fact that the distribution of each $\mathbf{z}_i$ is invariant to the permutation of its parents, we

will have that the permutation of $B$ will result into the same re-ordering of the $\mathbf{z}_i$'s i.e.:

$$\sigma(\mathbf{Z}_B) = g(\sigma(B)), \tag{2}$$

where $g(\cdot)$ is the function that maps $B$ to $\mathbf{Z}_B$. Finally, as each $y_i$ is a function, let it be $h(\cdot)$, of the specific $\mathbf{z}_i$, we will similarly have that $\sigma(\mathbf{y}_B) = h(\sigma(B))$. We have thus described that all of the aforementioned random variables are permutation equivariant to $B$ and as a result, due to the permutation invariant product / integral / summation operators, we will have that the FNP model is permutation invariant.

Continuing to the consistency under marginalization. Following [2] let us define $\tilde{\mathcal{D}} = \mathcal{D} \cup \{(\mathbf{x}_0, y_0)\}$ and consider two cases, one where the $\mathbf{x}_0$ belongs in $R$ and one where it doesn't. We will show that in both cases $\int p(\mathbf{y}_{\tilde{\mathcal{D}}}|\mathbf{X}_{\tilde{\mathcal{D}}})\mathrm{d}y_0 = p(\mathbf{y}_{\mathcal{D}}|\mathbf{X}_{\mathcal{D}})$. Lets consider the case when $\mathbf{x}_0 \in R$. In this case we have that the $M$ and $B$ sets will be the same across $\mathcal{D}$ and $\tilde{\mathcal{D}}$. As a result we can proceed as

$$\int p(\mathbf{y}_{\tilde{\mathcal{D}}}|\mathbf{X}_{\tilde{\mathcal{D}}})\mathrm{d}y_0 = \sum_{\mathbf{G},\mathbf{A}} \int p_\theta(\mathbf{U}_B|\mathbf{X}_B)p(\mathbf{G},\mathbf{A}|\mathbf{U}_B)p_\theta(\mathbf{Z}_B, \mathbf{y}_B|R, \mathbf{G}, \mathbf{A})\mathrm{d}\mathbf{U}_B\mathrm{d}\mathbf{Z}_B\mathrm{d}y_{i\in R\setminus\tilde{\mathcal{D}}_x}\mathrm{d}y_0. \tag{3}$$

Now we can notice that $y_{i\in R\setminus\tilde{\mathcal{D}}_x} \cup y_0 = y_{i\in R\setminus\mathcal{D}_x}$, hence the measure that we are integrating over above can be rewritten as

$$\int p(\mathbf{y}_{\tilde{\mathcal{D}}}|\mathbf{X}_{\tilde{\mathcal{D}}})\mathrm{d}y_0 = \sum_{\mathbf{G},\mathbf{A}} \int p_\theta(\mathbf{U}_B|\mathbf{X}_B)p(\mathbf{G},\mathbf{A}|\mathbf{U}_B)p_\theta(\mathbf{y}_B, \mathbf{Z}_B|R, \mathbf{G}, \mathbf{A})\mathrm{d}\mathbf{U}_B\mathrm{d}\mathbf{Z}_B\mathrm{d}y_{i\in R\setminus\mathcal{D}_x}, \tag{4}$$

where it is easy to see that we arrived at the same expression as the one provided at Eq. 8. Now we will consider the case where $\mathbf{x}_0 \notin R$. In this case we have that $R \setminus \tilde{\mathcal{D}}_x = R \setminus \mathcal{D}_x$ and thus

$$\int p(\mathbf{y}_{\tilde{\mathcal{D}}}|\mathbf{X}_{\tilde{\mathcal{D}}})\mathrm{d}y_0 = \sum_{\mathbf{G},\mathbf{A},\mathbf{a}_0} \int p_\theta(\mathbf{U}_B|\mathbf{X}_B)p(\mathbf{G},\mathbf{A}|\mathbf{U}_B)p_\theta(\mathbf{y}_B, \mathbf{Z}_B|R, \mathbf{G}, \mathbf{A})$$
$$p_\theta(\mathbf{u}_0|\mathbf{x}_0)p(\mathbf{a}_0|\mathbf{U}_R, \mathbf{u}_0)p_\theta(\mathbf{z}_0|\mathrm{par}_{\mathbf{a}_0}(R, \mathbf{y}_R))p_\theta(y_0|\mathbf{z}_0)\mathrm{d}\mathbf{U}_B\mathrm{d}\mathbf{Z}_B\mathrm{d}\mathbf{u}_0\mathrm{d}\mathbf{z}_0\mathrm{d}y_{i\in R\setminus\mathcal{D}_x}\mathrm{d}y_0. \tag{5}$$

Notice that in this case the new point that is added is a leaf in the dependency graph, hence it doesn't affect any of the points in $\mathcal{D}$. As a result we can easily marginalize it out sequentially

$$\int p(\mathbf{y}_{\tilde{\mathcal{D}}}|\mathbf{X}_{\tilde{\mathcal{D}}})\mathrm{d}y_0 = \sum_{\mathbf{G},\mathbf{A},\mathbf{a}_0} \int p_\theta(\mathbf{U}_B|\mathbf{X}_B)p(\mathbf{G},\mathbf{A}|\mathbf{U}_B)p_\theta(\mathbf{y}_B, \mathbf{Z}_B|R, \mathbf{G}, \mathbf{A})$$
$$p_\theta(\mathbf{u}_0|\mathbf{x}_0)p(\mathbf{a}_0|\mathbf{U}_R, \mathbf{u}_0)p_\theta(\mathbf{z}_0|\mathrm{par}_{\mathbf{a}_0}(R, \mathbf{y}_R))\left(\cancel{\int p_\theta(y_0|\mathbf{z}_0)\mathrm{d}y_0}^{\;1}\right)\mathrm{d}\mathbf{U}_B\mathrm{d}\mathbf{Z}_B\mathrm{d}\mathbf{u}_0\mathrm{d}\mathbf{z}_0\mathrm{d}y_{i\in R\setminus\mathcal{D}_x}. \tag{6}$$

$$= \sum_{\mathbf{G},\mathbf{A},\mathbf{a}_0} \int p_\theta(\mathbf{U}_B|\mathbf{X}_B)p(\mathbf{G},\mathbf{A}|\mathbf{U}_B)p_\theta(\mathbf{y}_B, \mathbf{Z}_B|R, \mathbf{G}, \mathbf{A})$$
$$p_\theta(\mathbf{u}_0|\mathbf{x}_0)p(\mathbf{a}_0|\mathbf{U}_R, \mathbf{u}_0)\left(\cancel{\int p_\theta(\mathbf{z}_0|\mathrm{par}_{\mathbf{a}_0}(R, \mathbf{y}_R))\mathrm{d}\mathbf{z}_0}^{\;1}\right)\mathrm{d}\mathbf{U}_B\mathrm{d}\mathbf{Z}_B\mathrm{d}\mathbf{u}_0\mathrm{d}y_{i\in R\setminus\mathcal{D}_x} \tag{7}$$

$$= \sum_{\mathbf{G},\mathbf{A}} \int p_\theta(\mathbf{U}_B|\mathbf{X}_B)p(\mathbf{G},\mathbf{A}|\mathbf{U}_B)p_\theta(\mathbf{y}_B, \mathbf{Z}_B|R, \mathbf{G}, \mathbf{A})$$
$$p_\theta(\mathbf{u}_0|\mathbf{x}_0)\left(\cancel{\sum_{\mathbf{a}_0} p(\mathbf{a}_0|\mathbf{U}_R, \mathbf{u}_0)}^{\;1}\right)\mathrm{d}\mathbf{U}_B\mathrm{d}\mathbf{Z}_B\mathrm{d}\mathbf{u}_0\mathrm{d}y_{i\in R\setminus\mathcal{D}_x} \tag{8}$$

$$= \sum_{\mathbf{G},\mathbf{A}} \int p_\theta(\mathbf{U}_B|\mathbf{X}_B)p(\mathbf{G},\mathbf{A}|\mathbf{U}_B)p_\theta(\mathbf{y}_B, \mathbf{Z}_B|R, \mathbf{G}, \mathbf{A})\left(\cancel{\int p(\mathbf{u}_0|\mathbf{x}_0)\mathrm{d}\mathbf{u}_0}^{\;1}\right)\mathrm{d}\mathbf{U}_B\mathrm{d}\mathbf{Z}_B\mathrm{d}y_{i\in R\setminus\mathcal{D}_x} \tag{9}$$

$$= \sum_{\mathbf{G},\mathbf{A}} \int p_\theta(\mathbf{U}_B|\mathbf{X}_B)p(\mathbf{G},\mathbf{A}|\mathbf{U}_B)p_\theta(\mathbf{y}_B, \mathbf{Z}_B|R, \mathbf{G}, \mathbf{A})\mathrm{d}\mathbf{U}_B\mathrm{d}\mathbf{Z}_B\mathrm{d}y_{i\in R\setminus\mathcal{D}_x} \tag{10}$$

where it is similarly easy to see that we arrived at Eq. 8. So we just showed that in both cases we have that $\int p(\mathbf{y}_{\tilde{\mathcal{D}}}|\mathbf{X}_{\tilde{\mathcal{D}}})\mathrm{d}y_0 = p(\mathbf{y}_{\mathcal{D}}|\mathbf{X}_{\mathcal{D}})$, hence the model is consistent under marginalization. $\qquad\square$

## D   Minibatch optimization of the bound of FNPs

As we mentioned in the main text, the objective of FNPs is amenable to minibatching where the size of the batch scales according to the reference set $R$. We will only describe the procedure for the FNP as the extension for FNP$^+$ is straightforward. Lets remind ourselves that the bound of FNPs can be expressed into two terms:

$$\mathcal{L} = \mathbb{E}_{q_\phi(\mathbf{Z}_R|\mathbf{X}_R)p_\theta(\mathbf{U}_R,\mathbf{G}|\mathbf{X}_R)}[\log p_\theta(\mathbf{y}_R,\mathbf{Z}_R|R,\mathbf{G}) - \log q_\phi(\mathbf{Z}_R|\mathbf{X}_R)]+$$
$$+ \mathbb{E}_{p_\theta(\mathbf{U}_{\mathcal{D}},\mathbf{A}|\mathbf{X}_{\mathcal{D}})q_\phi(\mathbf{Z}_M|\mathbf{X}_M)}[\log p_\theta(\mathbf{y}_M|\mathbf{Z}_M) + \log p_\theta(\mathbf{Z}_M|\mathrm{par}_\mathbf{A}(R,\mathbf{y}_R)) - \log q_\phi(\mathbf{Z}_M|\mathbf{X}_M)]$$
$$= \mathcal{L}_R + \mathcal{L}_{M|R}, \tag{11}$$

where we have a term that corresponds to the variational bound on the datapoints in $R$, $\mathcal{L}_R$, and a second term that corresponds to the bound on the points in $M$ when we condition on $R$, $\mathcal{L}_{M|R}$. While the $\mathcal{L}_R$ term of Eq. 11 cannot, in general, be decomposed to independent sums due to the DAG structure in $R$, the $\mathcal{L}_{M|R}$ term can; from the conditional i.i.d. nature of $M$ and the structure of the variational posterior we can express it as $M$ independent sums:

$$\mathcal{L}_{M|R} = \mathbb{E}_{p_\theta(\mathbf{U}_R|\mathbf{X}_R)}\left[\sum_{i=1}^{|M|}\mathbb{E}_{p_\theta(\mathbf{u}_i,\mathbf{A}_i|\mathbf{x}_i,\mathbf{U}_R)q_\phi(\mathbf{z}_i|\mathbf{x}_i)}\left[\log p_\theta\left(\mathbf{y}_i,\mathbf{z}_i|\mathrm{par}_{\mathbf{A}_i}(R,\mathbf{y}_R)\right) - \right.\right.$$
$$\left.\left. - \log q_\phi(\mathbf{z}_i|\mathbf{x}_i)\right]\right]. \tag{12}$$

We can now easily use a minibatch $\hat{M}$ of points from $M$ in order to approximate the inner sum and thus obtain unbiased estimates of the overall bound that depend on a minibatch $\{R, \hat{M}\}$:

$$\tilde{\mathcal{L}}_{M|R} = \mathbb{E}_{p_\theta(\mathbf{U}_R|\mathbf{X}_R)}\left[\frac{|M|}{|\hat{M}|}\sum_{i=1}^{|\hat{M}|}\mathbb{E}_{p_\theta(\mathbf{u}_i,\mathbf{A}_i|\mathbf{x}_i,\mathbf{U}_R)q_\phi(\mathbf{z}_i|\mathbf{x}_i)}\left[\log p_\theta\left(\mathbf{y}_i,\mathbf{z}_i|\mathrm{par}_{\mathbf{A}_i}(R,\mathbf{y}_R)\right) - \right.\right.$$
$$\left.\left. - \log q_\phi(\mathbf{z}_i|\mathbf{x}_i)\right]\right], \tag{13}$$

thus obtain the following unbiased estimate of the overall bound that depends on a minibatch $\{S, \hat{M}\}$

$$\mathcal{L} \approx \mathcal{L}_R + \tilde{\mathcal{L}}_{M|R}. \tag{14}$$

In practice, this might limit us to use relatively small reference sets as training can become relatively expensive; in this case an alternative would be to subsample also the reference set and just reweigh appropriately $\mathcal{L}_R$. This provides a biased gradient estimator but, after a limited set of experiments, it seems that it can work reasonably well.

## E   Predictive distribution of FNPs

Given the fact that the parameters of the model has been optimized, we are now seeking a way to do predictions for new unseen points. As we assumed that all of the reference points are a part of the observed dataset $\mathcal{D}$, every new point $\mathbf{x}^*$ will be a part of $O$. Furthermore, we will have that $B = \mathcal{D}_x = \mathbf{X}_{\mathcal{D}}$. We will only provide the derivation for the FNP model, since the extension to FNP$^+$ is straightforward. To derive the predictive distribution for this point we will rely on Bayes theorem and thus have:

$$p_\theta(y^*|\mathbf{x}^*,\mathbf{X}_{\mathcal{D}},\mathbf{y}_{\mathcal{D}}) = \frac{p_\theta(y^*,\mathbf{y}_{\mathcal{D}}|\mathbf{x}^*,\mathbf{X}_{\mathcal{D}})}{\int p_\theta(y^*,\mathbf{y}_{\mathcal{D}}|\mathbf{x}^*,\mathbf{X}_{\mathcal{D}})\mathrm{d}y^*}. \tag{15}$$

As we have established the consistency of FNP, we know that the denominator is $p_\theta(\mathbf{y}_{\mathcal{D}}|\mathbf{X}_{\mathcal{D}})$. Therefore we can expand the enumerator and rewrite Eq. 15 as

$$p_\theta(y^*|\mathbf{x}^*, \mathbf{X}_\mathcal{D}, \mathbf{y}_\mathcal{D}) = \sum_{\mathbf{G},\mathbf{A},\mathbf{a}^*} \int \frac{p_\theta(\mathbf{U}_\mathcal{D}|\mathbf{X}_\mathcal{D})p(\mathbf{G},\mathbf{A}|\mathbf{U}_\mathcal{D})p_\theta(\mathbf{Z}_\mathcal{D}, \mathbf{y}_\mathcal{D}|R, \mathbf{G}, \mathbf{A})}{p_\theta(\mathbf{y}_\mathcal{D}|\mathbf{X}_\mathcal{D})}$$
$$p_\theta(\mathbf{u}^*|\mathbf{x}^*)p(\mathbf{a}^*|\mathbf{U}_R, \mathbf{u}^*)p_\theta(\mathbf{z}^*|\mathrm{par}_{\mathbf{a}^*}(R, \mathbf{y}_R))p_\theta(y^*|\mathbf{z}^*)\mathrm{d}\mathbf{U}_\mathcal{D}\mathrm{d}\mathbf{u}^*\mathrm{d}\mathbf{Z}_\mathcal{D}\mathrm{d}\mathbf{z}^*,$$
$$(16)$$

where $\mathbf{a}^*$ is the binary vector that denotes which points from $R$ are the parents of the new point. We can now see that the top part is the posterior distribution of the latent variables of the model when we condition on $\mathcal{D}$. We can thus replace it with its variational approximation $p_\theta(\mathbf{U}_\mathcal{D}|\mathbf{X}_\mathcal{D})p(\mathbf{G},\mathbf{A}|\mathbf{U}_\mathcal{D})q_\phi(\mathbf{Z}_\mathcal{D}|\mathbf{X}_\mathcal{D})$ and obtain

$$p_\theta(y^*|\mathbf{x}^*, \mathbf{X}_\mathcal{D}, \mathbf{y}_\mathcal{D}) \approx \sum_{\mathbf{G},\mathbf{A},\mathbf{a}^*} \int p_\theta(\mathbf{U}_\mathcal{D}|\mathbf{X}_\mathcal{D})p(\mathbf{G},\mathbf{A}|\mathbf{U}_\mathcal{D})q_\phi(\mathbf{Z}_\mathcal{D}|\mathbf{X}_\mathcal{D})$$
$$p_\theta(\mathbf{u}^*|\mathbf{x}^*)p(\mathbf{a}^*|\mathbf{U}_R, \mathbf{u}^*)p_\theta(\mathbf{z}^*|\mathrm{par}_{\mathbf{a}^*}(R, \mathbf{y}_R))p_\theta(y^*|\mathbf{z}^*)\mathrm{d}\mathbf{U}_\mathcal{D}\mathrm{d}\mathbf{u}^*\mathrm{d}\mathbf{Z}_\mathcal{D}\mathrm{d}\mathbf{z}^*$$
$$(17)$$
$$= \sum_{\mathbf{a}^*} \int p_\theta(\mathbf{U}_R, \mathbf{u}^*|\mathbf{X}_R, \mathbf{x}^*)p(\mathbf{a}^*|\mathbf{U}_R, \mathbf{u}^*)p_\theta(\mathbf{z}^*|\mathrm{par}_{\mathbf{a}^*}(R, \mathbf{y}_R))$$
$$p_\theta(y^*|\mathbf{z}^*)\mathrm{d}\mathbf{U}_R\mathrm{d}\mathbf{u}^*\mathrm{d}\mathbf{z}^*$$
$$(18)$$

after integrating / summing over the latent variables that do not affect the distributions that are specific to the new point.