[Reviews · NeurIPS 2019]

Reviewer 1



Post-Rebuttal Comments: Thanks for the response, where the authors have the explained the posterior approximation and added additional results, I appreciate that. Also, I agree with the comments from other reviewers, that the proposed method still has several limitations and there is still room for improvement. Taking everything into consideration, I have raised my score by one. -------------Original Comments------------------ This paper proposes a new random function prior called functional neural processes (FNPs), and discussed approximate inference procedures. Empirical experimental analysis is also provided. The submission is well-written and very easy to follow, and I quite like the idea. *Significance*: 1, The FNP is significantly different from the previous NP formulation. The FNP improves the NP by replacing global latent variables in NP by the local ones. FNP combines local latent representations with graphical modeling that encode dependencies (which, in a sense, serves the role of global latent variable). 2, By this combination of NP and graphical modeling, FNP addresses some of the limitations of NP architecture. The original NP uses a finite-dimensional global latent variable and performs inference on this variable instead of functions, which does not construct a valid nonparametric prior. FNP removed this limitation and defines a nonparametric prior which could be potentially very useful. 3, Also, the original NP approach focuses more on meta-learning scenarios. On the contrary, FNP model enables learning in a more traditional regression/classification settings with improved uncertainty estimation. Overall, I believe FNPs could be a useful addition to BDL literature. *Weakness* While the high-level picture of this submission is solid, I have found that some of the details, especially the experimental evaluations, are not quite convincing. 1, A question regarding posterior derivation: I am not sure about the validity of equation (12). This essentially means the posterior distribution of test point only depends on the choice of inducing data, but not the value of training data. If the derivation is correct, then it indicates that under the approximation, FNP reduces to a parametric model and losses some advantage of being a "functional" prior. Am I missing something? Consider the sparse GP counterpart (which has a very similar graphical model of the relation between the inducing points and training points), all training data should also be used during prediction. 2, I appreciate the toy example part that demonstrates inductive biases, which is illustrative. I believe that FNP is expected to give better performance. However, we will have no idea how general those behaviors are unless more visualizable examples are included (even in the appendix) to thoroughly demonstrate this. It will also help rule out the randomness due to the choice of data-generating mechanism. 3, Furthermore, I have a few questions for the first task: i), how is the NP trained? The behavior of this NP is somewhat weird. If it is pre-trained on samples from RBF-GP, it is expected to behave like a GP, as demonstrated in the original NP paper. ii), How does FNP compare to other BNN inference algorithms? MC-dropout can often be very inaccurate in practice, therefore including other VI/MCMC BNNs could be more meaningful. 4, Since the contribution mainly comes from the architecture side, readers normally expect a thorough comparison to more BDL baselines other than dropout BNNs. Also, Apart from BDL literature, how does FNP perform compare to variational GPs and Deep GPs? According to the paper, FNP also uses GP kernel to carry some GP behaviors. I understand that FNP can be a useful non-GP alternative, and its non-GP behavior could be useful. However, this potential power has not been well-explored in the paper. Then, why not use GP and/or its extensions instead? 5, It seems that most experiments in this paper lack multiple runs and significance analysis. It is difficult to confirm the significance of FNPs for sure unless the mean and error bars from multiple runs are reported.

Reviewer 2



Summary: The authors propose a model for a stochastic process that learns a DAG whose nodes are the latent representations of each data point. A set of reference points R = (x_i)_{i=1}^K are used to anchor the distributions p(y_M|R,y_R,x_M) for a separate set of observations M. The authors claim better predictive performance and more robust uncertainty estimates compared to other baselines such as NPs. 1. A nice aspect of FNPs is their interpretability, in that for a given prediction, the bipartite graph given by A makes it clear which of the examples in R gave rise to a given prediction. 2. However my main point of concern is that the set R is fixed during training and evaluation, which to me seems to be a key difference between the FNP and the NP. I think this leaves the nice consistency property of p(y_B|x_B) (that didn’t hold for NPs) obsolete, you lose the flexibility of NPs in modelling a wide family of conditional distributions and hence the meta-learning(few-shot learning) application, and additionally makes the comparison with the NP unfair. If R is fixed, then in the end the FNP is limited to learning the conditional p(y_M|R,y_R,x_M) for a fixed R instead of learning the full distribution p(y_B|x_B) for arbitrary B. So the comparison against the NP is unfair in this case, since the NP has the much more difficult job of learning p(y_M|R,y_R,x_M) for arbitrary R. From line 262-263 I believe that for NP training you were selecting the context points randomly for each batch (this should be made clearer I think - how context & targets were chosen during NP training). For a fair comparison, you would need to either a) Train the FNP using different R’s for different batches or b) Train the NP keeping R fixed for each batch. I think getting a) to work would be much more interesting than seeing a comparison with b). Have you tried a)? 3. It seems like for the toy regression experiment, the NP was trained with context=target=full data (could this be made clearer in the text?). Note that this is different to how NPs are trained. Usually for NPs, you assume that you have access to multiple realisations of a stochastic process, and train on a random realisation for each training iteration. Here you are training on a single realisation, so using the NP in a setting that the NP wasn't made for, and it's been shown that if you train NPs this way for long enough on a fixed data set they just overfit to a deterministic function that goes through each data point - essentially the latent is ignored by the decoder and it just learns a mapping from x to y as in supervised learning (e.g. https://chrisorm.github.io/NGP.html). So I'm not sure how you got the NP to show high variance outside the training domain - did you not train till convergence? Regardless, it would be helpful to have more details on the NP training to clarify these matters. (Side note: however, it is still nice to see that FNPs can have more desirable behaviour for ood inputs (i.e. more like a GP) compared to NPs, whose behaviour with ood inputs is not well understood). 4. If R is fixed, I imagine that the predictive performance and quality of uncertainty is affected by the particular choice of R. This is also stated in the paper that quality of points in R mattered over quantity of points. Although there is an ablation study for different sizes of R, there doesn’t seem to be one for different choices of R given fixed size. I think it would be informative to offer guidelines in the choice of R, and if it’s random, it would be informative to have error bars for the numbers in Table1 across different choices of R. 5. I think the role of G and A in FNP is reminiscent of the roles of self-attention and cross-attention respectively in Attentive NPs (ANPs), where R is the context C and M is the target T. Both G and the self-attention in ANP learn which of the x_i in R are close to one another, and both A and cross-attention in ANP learn which of the x_i in R are close to the x_j in M. The difference is that in FNP discrete latent variables are used to parameterise “closeness” instead of deterministic real-valued attention weights in ANP. Another difference is that the y_R are predicted in FNP rather than fed into the encoder. In any case the FNP appears to be more closely connected to ANPs than NPs so perhaps it’s worth comparing to ANP as well? 6. One forward pass for FNPs will be O(|R|^2 + |R||M|) whereas it is O(|R| +|M|) for NPs. How does training time compare among the two? 7. for FNP+, it’s interesting that the model doesn’t ignore the z_i’s and predict y_i through just u_i, especially since u_i is computed directly from x_i. This leaves me thinking that not only is choosing the right tradeoff between dim(u) and dim(z) important, but the choice of mapping between x_i and u_i is also important - the richer this is, the more the model will want to use u_i exclusively and ignore z_i. If the model’s behaviour is sensitive to this choice, this could be a limitation of the model. Minor points: - Regarding the ordering of vectors in U_R, why that particular choice of t? Why is the property of u_ik > u_jk forall k => u_i > u_j desirable? And if so, why not use something simpler like t(u_i) = ||u_i||_2 ? - I think the notation for R,M and B can be a bit confusing because you define it to be a set of x’s but then use it as if they are a set of indices (e.g. U_R,X_B, i \in R). It would be clearer to just unify the notation and introduce the sets as sets of indices. Then it would be clearer to write (x_k,y_k)_{k \in par_Aj} instead of par_Aj(R,y_R). - Moreover I think the model dependencies would be slightly clearer if Equation (6) showed how the joint density p(y_B,z_B|R,G,A) decomposes rather than the marginal p(y_B|R,G,A), getting rid of the integrals. ***UPDATE: The rebuttal addresses some of the suggestions for improvement, so I've decided to increase my score by 1. However I still think that the focus of the paper should be on the FNP+randR case, and evaluation should also be done for a wider choice of R.

Reviewer 3



The paper indeed proposes and discusses two specific Functional Neural Process models, with detailed model formulation. Although the context is highly technical, I can basically follow the information flow which demonstrates certain level of the clarify. In some sense, the idea introduced in the paper is one step moving forward from the existing literature such as [14] and [36] by considering vectorial latent variables etc. It seems to me the model is extremely complicated so not sure how efficient it is in practice although the experiments show some comparable results while no efficiency performance has been shown.

[Author Response · NeurIPS 2019]

We would like to thank the reviewers for spending some of their time in thoroughly reviewing our work. Both R1 & R2 consider the work to be novel and significant, a fact that identifies the modelling framework of FNPs as a useful and important contribution to the community. The main negative points were in the experimental evaluation side and we will use this rebuttal as a way to address most of them.

The predictive distribution at eq. 12 is correct and a byproduct of the posterior approximations that we employed. If we instead employ the true posterior distribution over the latent variables, the predictive depends on the entire training dataset as $\mathbf{u}_R$ directly affects all of the points in $\mathcal{D}$ (see eq. 16 in the appendix). For this work we aimed for very simple posterior approximations and we left more involved ones as a point for future research. We further provide in the figure below an alternative toy regression fit with the same architectural details where we can see similar trends to the one provided in the main paper. All of the models were trained till convergence.

(a) MC-dropout    (b) Neural Process    (c) Gaussian Process    (d) FNP    (e) FNP$^+$

For all of the experiments in the paper, the NP was trained in a way that mimics the FNP, albeit we used a different set $R$ at every training iteration in order to conform to the standard NP training regime. More specifically, a random amount from $3$ to $num(R)$ points were selected as a context from each batch, with $num(R)$ being the maximum amount of points allocated for $R$. For the toy regression task we set $num(R) = N - 1$.

After the suggestions from the reviewers we also performed several additional experiments; we trained an NP with the same fixed reference set $R$ as the FNPs throughout training, a standard variational Bayesian neural network with Gaussian priors / approx. posteriors over the weights and an FNP+ where we randomly sample a new $R$ for every batch (akin to the NP and) and use the same $R$ as the NP for evaluation. The results from these models can be seen at the table below and, as we can see, the FNPs still provide robust uncertainty while the randomness in $R$ usually improves the o.o.d. detection, possibly due to the implicit regularization. We also included the results (mean & standard error) from both FNPs obtained after 5 replications with different $R$s of $num(R) = 300$ (the one used in the paper); the error bars are larger on CIFAR 10 than MNIST, possibly due to it being a harder task. It should also be mentioned that the primary motivation for FNPs was not meta-learning, although it is something we plan to explore for future work.

|  | VI BNN | NP fix$R$ | FNP+ rand$R$ | FNP 5repl. | FNP+ 5repl. |
|---|---|---|---|---|---|
| MNIST | 0.02 / 0.6 | 0.01 / 0.6 | 0.02 / 0.8 | 0.02± 0.0 / 0.7±0.0 | 0.02±0.0 / 0.7±0.0 |
| nMNIST | 1.33 / 99.80 | 1.09 / 99.78 | 2.20 / 100.0 | 1.95±0.06 / 99.93±0.03 | 1.97±0.05 / 99.97± 0.02 |
| fMNIST | 0.92 / 98.61 | 0.64 / 98.34 | 1.58 / 99.78 | 1.69±0.05 / 99.43±0.10 | 1.63±0.04 / 99.58±0.07 |
| Omniglot | 1.61 / 99.91 | 0.79 / 99.53 | 2.06 / 99.99 | 1.88±0.04 / 99.86±0.04 | 1.85±0.06 / 99.90±0.03 |
| Gaussian | 1.77 / 100.0 | 1.79 / 99.96 | 2.28 / 100.0 | 1.95±0.14 / 99.81±0.16 | 2.07±0.02 / 99.98±0.02 |
| Uniform | 1.41 / 99.87 | 1.42 / 99.93 | 2.23 / 100.0 | 1.99±0.06 / 99.96±0.02 | 1.95±0.06 / 99.96±0.02 |
| CIFAR10 | 0.06 / 6.4 | 0.07 / 7.5 | 0.09 / 6.9 | 0.17±0.01 / 7.5±0.08 | 0.08±0.01 / 7.3±0.04 |
| SVHN | 0.45 / 91.8 | 0.46 / 91.5 | 0.56 / 91.4 | 0.86±0.05 / 90.74±0.81 | 0.51±0.04 / 91.3±0.76 |
| tImag32 | 0.52 / 91.9 | 0.55 / 91.5 | 0.77 / 93.4 | 1.22±0.02 / 94.49±0.29 | 0.69±0.02 / 92.6±0.39 |
| iSUN | 0.57 / 93.2 | 0.60 / 92.6 | 0.83 / 94.0 | 1.33±0.02 / 95.71±0.24 | 0.75±0.02 / 93.8±0.38 |
| Gaussian | 0.76 / 96.9 | 0.20 / 87.2 | 1.23 / 99.1 | 1.05±0.10 / 93.73±1.29 | 0.60±0.08 / 93.6±1.09 |
| Uniform | 0.65 / 96.1 | 0.53 / 94.3 | 0.90 / 97.2 | 0.85±0.16 / 89.43±4.20 | 0.61±0.11 / 93.4±1.89 |

As for the comparison to variational / deep GPs; unfortunately, due to lack of time we cannot perform experiments with these baselines. On a theoretical level, the FNPs can be more scalable due to not having to invert a matrix for prediction. Furthermore, they can easily support arbitrary likelihood / noise models (e.g. for discrete data) in a straightforward way, in contrast to GPs where we have to consider appropriate transformations / warpings of a Gaussian distribution that usually require further approximations.

Similarities with ANP; indeed we did mention it in the related work section. We can view the attention weights of ANP as providing the graph edge probabilities in FNP. The key difference between the FNP and NP still applies for the ANP though, as ANP still has a global latent variable. Unfortunately, due to lack of time we cannot perform experiments with ANP, but we believe that the additional results we presented in this rebuttal make FNP contribution convincing.

Mapping for $\mathbf{u}$; indeed the type of this mapping is important and the reason why appropriate regularization is necessary. This is also one of the reasons why we tied most of the model / variational parameters, as mentioned in section 2.2. It should be mentioned that NPs can have similar drawbacks, e.g. how do you select the appropriate way to incorporate the global context $\theta$ in the prediction. Having said that, we can also view it as a capability of the FNPs, since it can serve as a knob that can be adjusted for the specifics of a given application.

[Meta-Review · NeurIPS 2019]

This is a solid submission providing interesting insights and methodological advances. While the reviewers had some concerns, most of them were clarified with the rebuttal. The reviewers found the additional results informative and I encourage the authors to include them in the final version of the paper. I would also encourage the authors to discuss the drawbacks of the proposed approach in more detail.